# Contribution of the HIV-1 Envelope Glycoprotein to AIDS Pathogenesis and Clinical Progression

**DOI:** 10.3390/biomedicines10092172

**Published:** 2022-09-02

**Authors:** Agustín Valenzuela-Fernández, Romina Cabrera-Rodríguez, Concha Casado, Silvia Pérez-Yanes, María Pernas, Jonay García-Luis, Silvia Marfil, Isabel Olivares, Judith Estévez-Herrera, Rodrigo Trujillo-González, Julià Blanco, Cecilio Lopez-Galindez

**Affiliations:** 1Laboratorio de Inmunología Celular y Viral, Unidad de Farmacología, Sección de Medicina, Facultad de Ciencias de la Salud, Universidad de La Laguna (ULL), 38071 La Laguna, Spain; 2Unidad de Virología Molecular. LRIR, Centro Nacional de Microbiología (CNM), Instituto de Salud Carlos III, 28220 Madrid, Spain; 3AIDS Research Institute IrsiCaixa, 08916 Badalona, Spain; 4Analysis Department, Faculty of Mathematics, Universidad de La Laguna (ULL), 38296 La Laguna, Spain; 5Institut de Recerca en Ciències de la Salut Germans Trias i Pujol (IGTP), 08916 Badalona, Spain; 6Chair of Infectious Diseases and Immunity, Faculty of Medicine, University of Vic-Central University of Catalonia (UVic-UCC), 08500 Vic, Spain; 7CIBERINFEC, ISCIII, 28029 Madrid, Spain

**Keywords:** HIV-1 Env function, natural control of the infection, elite controllers

## Abstract

In the absence of antiviral therapy, HIV-1 infection progresses to a wide spectrum of clinical manifestations that are the result of an entangled contribution of host, immune and viral factors. The contribution of these factors is not completely established. Several investigations have described the involvement of the immune system in the viral control. In addition, distinct HLA-B alleles, HLA-B27, -B57-58, were associated with infection control. The combination of these elements and antiviral host restriction factors results in different clinical outcomes. The role of the viral proteins in HIV-1 infection has been, however, less investigated. We will review contributions dedicated to the pathogenesis of HIV-1 infection focusing on studies identifying the function of the viral envelope glycoprotein (Env) in the clinical progression because of its essential role in the initial events of the virus life-cycle. Some analysis showed that inefficient viral Envs were dominant in non-progressor individuals. These poorly-functional viral proteins resulted in lower cellular activation, viral replication and minor viral loads. This limited viral antigenic production allows a better immune response and a lower immune exhaustion. Thus, the properties of HIV-1 Env are significant in the clinical outcome of the HIV-1 infection and AIDS pathogenesis.

## 1. Introduction

In the absence of antiretroviral therapy (ART), the hallmark of human immunodeficiency virus type 1 (HIV-1) infection is the gradual destruction of the naive and memory CD4+ T-lymphocytes and the associated immunological abnormalities leading to the acquired immunodeficiency syndrome (AIDS) [1,2,3,4,5,6,7,8,9,10,11] (Figure 1, progressors immune system damage, top box). The severity of the symptoms and viral transmission strongly correlate with the peak of viral load (VL) during primary infection and the subsequent viral set-point [12,13,14,15,16,17,18,19,20,21,22,23,24]. HIV-1 infection is characterized by a wide spectrum of disease outcomes according to the progression time of patients. Different nomenclatures have been used to name the distinct groups of HIV-1 individuals (reviewed in [25]). The typical HIV-1 infected patient, in the absence of ART, progresses to AIDS and death over a period of about 8–10 years after seroconversion [5,26,27]. Some patients, designated rapid progressors (RPs), progress to AIDS within three years of primary infection [26,28,29]. On the other side, there is a small subset of HIV-1 individuals that are able to permanently control viral replication and clinical progression and might never progress or progress very slowly [5,30] (Figure 1, non-progressors immune system damage, bottom box). In general, these subjects have been infected with HIV-1 for more than ten years, maintaining high CD4+ lymphocyte numbers, undetectable VL, without clinical symptoms, and remaining therapy naïve [31]. These individuals have been defined as long-term non-progressors (LTNPs) [32,33,34,35,36], elite controllers (ECs) [30,37], slow progressors [38,39,40], HIV controllers (HICs) [41] and elite suppressors [42]. Within this set of individuals, some subgroups can be distinguished in terms of VL: viremic LTNPs or viremic controllers (vLTNPs or LTNP-VCs) with VLs between 50 and 2000 copies/mL [32], LTNPs viremic non-controllers (LTNP-NCs) with VLs above 2000 copies/mL [43], and LTNP-Elite controllers (LTNP-ECs) with undetectable VLs (<50 copies/mL) [26]. In the LTNP-ECs subgroup, there is a natural control of the infection without any ART, maintaining undetectable HIV-1 VLs for long periods of time (even for more than 20–30 years) and lack of clinical progression [44]. This clinical phenotype is the consequence of the necessary cooperative interaction of host, immune, and viral factors [26,30,44,45,46,47,48,49,50,51,52,53,54,55,56,57,58,59,60,61,62,63,64,65,66] (Figure 1, non-progressors bottom box). Several investigations described the contribution of the immune system, both at the cellular and serological level, in the primary and the subsequent control of the viral infection. This control is the result of many elements and the activity of different cell types, such as CD4+ and CD8+ T cells, natural killers (NKs), dendritic cells (DCs), different types of antibodies (Abs), cell restrictions factors, human leucocyte antigens (HLAs) genotypes and/or host factors like CCR5 protective mutations [26,44,45,48,49,50,51,52,53,54,56,61,66,67,68,69,70,71,72,73], as summarized in Figure 1. In addition, HLA-B genotypes HLA-B57/B58 or -B27 [63], HLA-B*35:01 [74,75] and HLA-C [26,76,77], such as the HLA-C*03:02 1 in an African Pediatric Population [78], are linked with the control of HIV-1 infection (Figure 1, non-progressors bottom box). In some LTNP individuals [79] that harbor viruses with low replication capacity [80,81,82,83], the HIV-1 LTNP phenotype has been associated with the presence of potent and broad cytotoxic T lymphocyte (CTL) responses [66,84] (Figure 1, non-progressors bottom box) and active NK cells.

The role of the virus and viral proteins has been less investigated perhaps because in a limited, but influential study, replicating viruses were isolated from a group of non-progressor individuals [85]. However, important deletions in the HIV-1 *nef* gene have been identified in viruses isolated from a cohort of Australian LTNPs [86], and many mutations in other genes were reported [79] in non-progressor individuals. Among the HIV- 1 viral proteins, the reverse transcriptase (RT), because of its essential role in replication and because it is the main target of ART, has been extensively studied [87,88,89,90,91,92,93,94,95,96,97,98]. In addition, the proteins of the *gag* gene are involved in the virion structure and also in the cellular immune response [99,100,101,102,103,104,105,106,107]. Apart from these proteins, the viral envelope gene (*env*) and the derived glycoprotein complex (Env) have attracted numerous studies because of their central role in (i) the initial events (CD4 and co-receptor binding, pore fusion formation) and subsequent step of the biological viral life cycle; (ii) the neutralizing humoral immune response; and (iii) the viral tropism. The results of several studies on the viral Env/*env* investigating their role in viral pathogenesis will be summarized in this review.

## 2. HIV-1 Envs from LTNP-EC Individuals Present Inefficient Viral Functions, Associated with the Natural Control of the Infection and the Non-Progressor Clinical Phenotype

The investigation of the HIV-1 *env*/Env functions was undertaken analyzing viruses from HIV-1 individuals with different clinical phenotypes: LTNP-ECs, viremic non-progressors, progressors and rapid progressors (RPs) [26,44,70,71,72,73,83,121,122,123,124,125,126,127]. In our initial studies, we focused on LTNP individuals infected for long periods of time (i.e., more than 10 years and with more than 25–30 years of clinical follow-up) [26,44,70,72,121,127]. The isolated viral *env* sequences (full-length viral *env*) from infected individuals were cloned into expression plasmids (Figure 2a). The viral clones were completely sequenced at the nucleotide level and submitted to phylogenetic analysis (Figure 2a). These viral Envs were then characterized by multiple phenotypic test/assays to disclose the principal properties of their viruses (Figure 2b–e). In order to do this, we developed several techniques to study the functions of viral Env during the first steps of the viral cycle. This phase of the viral cycle is a complex multistage process with highly regulated steps involving many cellular molecules mobilized by the Env viral binding to CD4, CXCR4 or CCR5 receptors [128,129,130,131,132,133,134,135,136,137,138,139]. As we and others described, these interactions end in the formation of a fusion pore through which the viral capsid enters the cell [123,125,140,141,142]. The efficient pore fusion formation relies on key signals triggered by Env-CD4 interaction promoting cytoskeleton modifications [123,125,140,142], such as microtubules (MTs) acetylation in the α-tubulin subunit [142], F-actin severing, and capping reorganization [123,125]. A deficiency in these HIV-1-Env-mediated signals leads to a defect in the early steps of viral infection and replication [123,125,140,141,143,144,145], which ends in a limited viral replication.

Based on these experimental strategies, we reported that Envs from viruses of a cluster of LTNP-EC individuals showed a limited ability to induce both cortical F-actin reorganization and capping, and a low signal stabilizing acetylated MTs in the α-tubulin subunit [121] (Figure 3). These HIV-1 Envs were well expressed at the cell-surface of virus producing cells (Figure 2b). These data correlated with the fact that these Envs were not able to bind to CD4 with high affinity [70,71,72,121,123,125,140,142] (Figure 2c–e and Figure 3). The inability of these ECs’ HIV-1 Envs to bind to CD4 and trigger cell signals to reorganize and modify the cytoskeleton, to generate a pseudopod where cytoskeleton and cell-surface receptors for HIV-1 infection concentrate [121,123,125] (Figure 3), accounts for the defect of the HIV-1 Envs for the promotion of pore fusion formation and transfer of viral material to primary non-infected CD4+ T cells [121] (Figure 2c,d and Figure 3). In addition, a significant correlation was observed between the HIV-1 transfer values, mediated by Env/CD4 binding capacity and fusogenicity, thus linking the fusion defect to a low CD4 affinity (Figure 2c,d). Before characterizing these mechanisms underlying inefficient HIV-1 Envs, we reported non-functional viral Envs in a previous study with non-progressor individuals [26,73,83]. Taken together, these data further confirm the deficient Env fusion capacity observed in the cluster LTNP envelopes, correlating with the inefficiency to infect target cells and replicate at high levels [26,121] (Figure 2e and Figure 3).

It is assumed that functional viruses from HIV-1 infected patients can be selectively cleared by the immune response. In the case of LTNP-ECs, these functional viruses are not entirely removed from the organism due to their chromosomal integration in silent areas, conferring deep latency [120]. Likewise, HLA-B*27, HLA-B*57 and HLA-B*14 alleles appear to be prevalent among LTNP-EC individuals and have been proposed to be responsible for the observed virological control [61,62,146] (Figure 1, non-progressors bottom box). All participants in our studies presented diverse HLA-B protective alleles [70,121]. Moreover, all of them were infected with HIV-1 viruses bearing non-functional Envs, being indicative of the deficient Env functions being associated with low or undetectable viremia and allowing the virological control in these individuals (Figure 1, non-progressors bottom box). 

The heritability of HIV-1 virulence has been reported in large cohorts of viruses or in transmission pairs [142]. Studying an HIV-1 LTNP-ECs viral cluster, we reported for the first time that the viral Envs’ characteristics were inherited by all the viruses of the cluster, independently of the patients’ host markers [121]. The heritability of the inefficient functionality of the viral Env resulted in the same specific non-evolving clinical phenotype in these ECs [121]. These Envs could represent interesting immunogenic viral *env*/Env tools for vaccine programs.

HIV-1 infection is characterized by a high-level viral replication and elevated mutation rates, quantified at about one mutation every replication cycle [147]. This mutation rate leads to an extensive virus variation which has been estimated in an *env* gene sequence at 0.6–1% substitutions per year within infected patients [148,149,150,151]. Furthermore, when comparing HIV-1 *env* sequences from patients infected by different HIV-1 subtypes, they could differ on average by 25% and by as much as 35% [150,152], and this variation could reach a 1–2% of genetic variation within a transmission pair or within a single individual. Furthermore, HIV-1 Env shows extensive conformational adaptability and massive glycan shielding (i.e., Env glycans accounts for about 1/2 the mass of the molecule) that allow the virus to evade the effects of neutralizing antibodies (nAbs) and other viral antigen-triggered immune responses [46,47,150,151,153,154,155,156,157,158,159,160,161]. It is thought that transmitted HIV-1 tends to display Env proteins with less glycosylation than those that are poorly transmitted [162,163,164]. This HIV-1 *env*/Env diversity poses major challenges for the development of preventive vaccines. 

In our work studying viral Envs form viruses of LTNP-ECs, we reported by phylogenetic analysis that the nucleotide sequences of the *env* genes from LTNP-ECs [26,121] had a very limited evolution in comparison with other viral Envs from contemporary viruses from Europe and North America [121]. These characteristics of the *env* sequences of the EC-cluster viruses are indicative of a very limited number of replication cycles and that the functional defects could be conserved, impairing replication capacity and transmissibility [121]. In fact, after analyzing these ECs’ sequences, we identified three mutations in the V1, C2, and V4 *env* regions, as follows: positions 140 (T140, V1), 279 (A279, C2) and 400 (T400, V4), respectively. These *env* mutations were associated with inefficient signaling, fusion and infection of the Env of this cluster of viruses [121]. It is important to highlight that the viral samples in these subjects were taken at least 15–20 years after the primary infection of the LTNP-EC individuals.

The sequences from the LTNP-ECs showed an “old” viral dating [165], with a very short distance to MRCA (most recent common ancestor of the HIV-1 subtype B reference nucleotide sequence) signifying the limited viral evolution in these LTNP-ECs [70]. In the analysis of sequences, the variable loops of the gp120 subunit of the Env (i.e., V2, V4, and mainly V5) of the controller subjects showed shorter and less glycosylated sequences than in progressor individuals [70]. The lack of viral evolution in these LTNP-EC Envs [70] and in previous studies with viruses from different LTNP-ECs [121] support the notion that the viruses from LTNP-ECs are very close to the Transmitted/Founder (T/F) viruses (Figure 1, non-progressors bottom box). These viral LTNP-EC Envs were well expressed at the plasma membranes of virus-producing cells (Figure 2b) but present inefficient HIV-1 Env-mediated cell-to-cell fusion (pore fusion formation assay) and viral transfer to primary non-infected CD4+ T cells from healthy donors [70] (Figure 2c,d), both events driven by the first HIV-1 Env/CD4 interaction which correlates with a very poor viral evolution and diversity. These facts are related to the poor affinity for CD4 shown by these EC Envs and without any particular mutation pattern in the viral Env [70]. Taken together, these results indicate that deficient Env viral functions and shorter *env* sequences are associated with viral control and a low clinical progression rate in HIV-1 LTNP-ECs. This observation points to the role of genotypic and phenotypic Env characteristics in the extent of HIV-1 replication *in vivo* and in its related pathogenesis.

## 3. Fully Functional HIV-1 Envs Are Linked to Viremia and Progressor Clinical Phenotypes

For the investigation into the role of viral Env in the control of HIV-1 infection and pathogenesis, we also analyzed viral *envs*/Envs from other sets of viruses from non-clustered LTNP-EC individuals, followed for more than ten years, in comparison with viruses from patients infected at the same period of time but with progressor phenotypes [70].

In contrast to the Envs from the LTNP-EC subjects, the viruses from the progressor individuals (viremic and progressors) showed the opposite properties, with a good affinity for CD4, cell fusion and viral transfer [70,72,121,127]. Therefore, functional HIV-1 Envs are associated with infectious virus and cytopathic activity [71,72], which characterize viremic and progressor/RP clinical outcomes [70,72,121,127] (Figure 3 and Figure 4). Functional HIV-1 Envs favor the accumulation of mutations that could result in function gains of the Envs and the evasion from immune responses (Figure 3 and Figure 4). We observed this pattern in the HIV-1 Envs, from more recently infected individuals and vLTNPs [70]. Looking for potential factors associated with this increase in viral infectivity, we reported that an increase in Env functionality correlates with longer and more glycosylated proteins. This conclusion arose after studying the protein sequences from the viruses from individuals with different clinical groups focusing on the variable loops and their associated potential N-linked glycosylation sites (PNGs) in the gp120 subunit of the Env [70]. There is a trend in the HIV-1 viral Env to gain length and glycosylation sites along the epidemic [166,167,168]. This increasing trend is also found in our work where viruses from the LTNPs (EC and viremic) and progressors Envs isolated in the 90s showed shorter dimensions than those of the progressors group obtained more recently (2013–2014) [70]. We observed that Env changes accumulated essentially in the V1, V2, V4 and V5 loops [70]. The increase in the length and PNGs of the V1–V2 region has been reported through chronic infections from early to late viral Env sampling [167] as we reported [70]. Likewise, works relating the role of V1 and V4 loops in the CD4 binding neutralization observed a similar change in these loops [169,170,171,172], and in the viral cell-to-cell transfer capacity [168,173,174], as we reported [70]. It is worth noting that the V3 loop presents a complete stability in length and glycosylation sites. This V3 loop is key for viral tropism [175,176,177,178,179] and for the correct CD4 Env binding as revealed with anti-V3 neutralizing antibodies that abrogate Env-CD4 interaction, as the authors and others have reported [180,181]. Furthermore, during chronic infection, higher PNG density has been observed in the V1–V5 region of the gp120 subunit of the HIV-1 Env complex compared to the PNG density observed during the early acute phase of the infection [182]. 

During viral transmission to a new host, a selection for viral variants with shorter variable regions and a reduced degree of PNGs occurs in viruses from the HIV-1 subtype B [183]. An increase in viral infectivity and replication capacity has been associated with genetic variability in the *env* gene [184,185,186,187,188,189]. This viral replication could favor the gain of function of the HIV-1 *env* by increasing viral fitness and could result in the escape from the immune response and ART [190,191,192,193,194,195,196,197,198]. In our studies, we detected the loss of the N362 PNGs (*HXB2 isolate*; *group M*, *subtype B* (*HIV-1 M:B_HXB2R:NCBI:txid11706*)), which is frequently observed in the Envs of non-progressor phenotypes (EC and viremic patients) and in long-lasting progressors, but not in HIV-1 Envs from more recent progressors. This change could be related to a gain of functionality observed in these Envs [70]. However, in a study of some Australian viruses presenting the N362 glycosylation site, the viruses showed efficient fusion and transfer capacity [199]. These data reflect the significant effect that point mutations could have in the viral characteristics and HIV pathogenesis [70,121,200,201]. These results indicate that deficient viral Envs are associated with non-progressor, controller individuals and that fully functional HIV-1 Envs are mainly linked to viremic and progressor clinical phenotypes.

## 4. Role of the Viral Env Complex in Signal Transmission in Other Cellular Process and Cell Death

In addition to these direct viral effects on viral replication, the Env complex is also associated with other important cellular processes like fusion pore formation and autophagy/cell death dysregulation. These processes are summarized in Figure 3. In the characterization of these HIV-1 Envs properties, we analyzed HIV-1 Envs from viremic non-progressors (VNPs), progressors and rapid progressors (RPs) patients [44,70,72,73,121,127]. The phenotypic characterization of the Envs from HIV-1 progressors (Figure 3 and Figure 4) indicated higher replication capacity for these HIV-1 viruses when compared with HIV-1 Envs of viruses from the LTNP-EC cluster [121]. HIV-1 Envs from progressors patients are associated with functional Env showing efficient CD4 binding and signaling that promote cytoskeleton reorganization and the formation of the pseudopod-hot region. This process allows an efficient HIV-1 infection (Figure 3a) which leads to higher fusogenic, viral transfer and infection capacities than viruses from LTNP-ECs [70,121]. It is noteworthy that we reported that the HIV-1 gp41-Env subunit promotes bystander cell-death by autophagy and apoptosis [114,115,116,117,118,119]. Our works indicate that functional HIV-1 Envs from VNPs and RPs promote bystander cell death in uninfected CD4+ T cells by triggering late autophagy [71,72] (Figure 3b). In line with this Env functionality, we found significant *env* gene diversity in sequences isolated from VNPs compared with RPs, correlating with the efficient ability of these VNP HIV-1 Envs to infect and favor virus replication [72]. A similar observation has been reported indicating that viral population diversity remains higher in VNPs compared to standard progressors or RPs [202]. Persistent HIV-1 replication in the presence of supposedly efficient immune responses in VNPs is expected to lead to the accumulation of mutations to compensate viral fitness cost, which could result in a continuous Env escape from neutralizing Abs [203,204]. Furthermore, in the late AIDS phase of chronic infection in RPs, uncontrolled HIV-1 replication occurs together with the selection of the fittest variants [148]. Therefore, functional HIV-1 Envs are directly associated with infectious viruses of viremic and progressors patients in which HIV-1 infection evolves [70,71,72,127] (Figure 1, Figure 3 and Figure 4).

This efficient viral function of the HIV-1 Env, in a CD4 dependent manner, allows the virus to overcome cell barriers that limit HIV-1 Env-mediated pore fusion formation, viral entry to the cell, infection and replication (Figure 3a). A key restriction factor for HIV-1 infection that we characterized is the cytoplasmic enzyme HDAC6 (histone deacetylase 6) [70,71,72,126,127,142,205], and more recently the transactive response of the DNA-binding protein (TARDBP or TDP-43) together with HDAC6 (i.e., the TDP-43/HDAC6 axis) [127]. An increase in the expression of functional TDP-43 concomitantly enhances the levels of mRNA and protein of HDAC6, leading to a diminution of the activity of functional HIV-1 Envs from viruses of VNP and RP individuals, reaching the levels of the inefficient Env from LTNP-EC individuals [127]. Silencing of the endogenous TDP-43 strongly reduces the levels of mRNA and of the HDAC6 enzyme [127], stabilizing acetylated MTs that favor the infection activity of primary HIV-1 Envs of VNP, progressors and even of non-functional Envs from LTNP-EC individuals [127]. This last observation suggests that defective viral features observed in a virus of LTNP-ECs [70,71,121,206] are possibly also modulated by the TDP-43/HDAC6 axis [127]. The TDP-43/HDAC6 axis therefore regulates cell permissivity to HIV-1 infection. This point may have negative consequences in HIV-1 LTNP-EC individuals, particularly if a negative regulation of TDP-43 occurs with a concomitant decrease in HDAC6 that would render cells more permissive against inefficient LTNP-EC Envs. Consistently, it has been reported that the ability of the viral Env to trigger signals that overcome the HDAC6 barrier is directly related to its fusion and infection activities [70,72,121,126,142]. The TDP-43/HDAC6 axis could be another factor that is worth exploring in EC individuals that lose the natural control of the infection [70,72,121,126,142,207]. Taken together, these data support the fact that functional HIV-1 Envs are associated with progressors’ clinical outcomes.

## 5. Discussion

It is thought that the host-immune response is the main factor responsible for viral control in HIV-1 patients [30,208], restraining viral replication and functionally clearing viruses from the organism. Moreover, HLA-B*27, HLA-B*57 and HLA-B*14 alleles have been associated with enhanced virological control and are prevalent among ECs [61,62,146]. In our studies, all the characterized HIV-1 LTNPs individuals presented different HLA-B alleles, even among clustered patients. Likewise, these patients did not present host immune factors associated with infection protection and/or low disease progression, such as CCR5 mutations, etc. [26,44,70,72,73,121,127]. Furthermore, our group and others reported a direct relationship between HIV-1 Env functional deficiencies and long-term viremia control in LTNP-EC individuals [70,80,121,127]. 

We have also determined, to the best of our knowledge for the first time, that a defect in the Env binding to CD4 and the triggered signal led to low fusion, infection and replication capacities, and transmissibility of viruses from these LTNP-EC individuals. The Envs from these viruses were ineffective in the CD4 binding and in the associated functions: viral signaling, fusion, cell entry and infection. The inefficient functionality of the glycoprotein in LTNPs and LTNP-ECs was not a consequence only of the immune response. The deficient Env characteristics determine the low replication and transmissibility of ECs’ viruses [70,72,121,125,126,127,142], explaining the clinical outcome (i.e., non-progression) and the LTNP-EC phenotype [70,121,127] (Figure 4). It is conceivable that deficient Envs of viruses of LTNPs (i.e., viremic and ECs) and their low variability would allow the immune system to better control the HIV-1 infection (Figure 4). 

Viral Envs from LTNPs exhibited non-functional characteristics in comparison with those from viruses of the progressive infection groups, supporting the view that the properties of the Envs directly condition HIV-1 infection. Thus, poor viral Env functions correlate with viral control and low clinical progression rate in HIV-1 individuals, whereas functional Envs are linked to viruses of patients lacking viral control and presenting clinical progression [44,70,72,73,80,121,127] (Figure 4). These results point to the important role of HIV-1 Env characteristics in determining patients’ clinical outcome (Figure 4), and that these properties were inherited in every virus of a well-defined LTNP-EC cluster [121]. These data support the role of the deficient HIV-1 Envs in the LTNP-EC phenotype which lacks viral pathogenesis.

In the studies reviewed here, our group and other teams reported deficient functions of HIV-1 Envs from LTNP-ECs, in clustered [80,121] as well as in non-clustered HIV-1 individuals controlling viremia [70,127]. On the other hand, a progressive increase in functionality of Env was found in viruses from LTNPs to chronic no controller HIV-1 patients [70]. An increase in the PNGs and in the length of these loops was associated with this functional improvement, which was observed in every Env characteristic studied: fusion, virus transfer and infection [70]. This gain of functionality in the HIV-1 Envs was supported by different strategies, co-receptor binding, neutralization or tropism [167]. It is conceivable that better HIV-1 Env functions could promote viral fitness and would therefore favour resistance against the immune response [190,191,192,193,194,195,196,197,198,209]. It is noteworthy that HIV-1 *env* variability has also been related with better viral infectivity and replication capacities [184,185,186,187,188,189]. A last example was reported in a LTNP-EC who discontinued ART [210]. In this case, the Env-V1 region presented two additional PNGs while conserving the infection and replicative capacities. In fact, after analyzing 6,112 Env/*env* sequences deposited in the Los Alamos National Laboratory online database, this unusual Env/*env* sequence ranked in the top 1% of length [210].

In a different study, we reported that functional HIV-1 Envs are associated with viruses of viremic and progressors (RPs and Progressors) HIV-1 patients. In the HIV-1 *env* sequences isolated from VNPs and RPs, we detected a significantly higher *env* gene diversity in VNPs compared to Progressors or RPs [202]. We described that HIV-1 Envs of VNPs, Progressors and RPs patients are equally functional for CD4-mediated pore fusion formation, viral transmission, virus entry and infection, and for productive signaling as well [70,72,121,127]. Furthermore, it seems that Env plasticity allows for the maintenance of proper fusogenic function in both HIV-1 Envs from VNP and RP individuals, despite representing apparent opposite clinical situations for HIV-1 replication *in vivo* (i.e., viremic “non-progressors” vs. RPs) [72]. We showed for the first time, to the best of our knowledge, that viral R5-tropic Envs isolated from VNPs and RPs promote (a later) autophagy in uninfected CD4+ T cells [72]. These Envs stimulated cell death by the contact of bystander non-infected cells that strongly correlated with the fusogenic capacity of these functional Envs [72]. We found that several Env functions were comparable between virus of VNPs and RPs patients, implying that HIV-1 Env function, at least in these patients, does not have a major role in the VNP phenotype [72]. Hence, some other immune mechanism must be responsible for this viremic non-progressor clinical phenotype, such as lower CCR5 expression, lower immune cells activation [211] and/or the predominant existence of long-lived central memory CD4+ T cells [211]. However, the efficiency of VNPs HV-1 Env-in viral replications should keep us alert for an eventual AIDS development when left untreated [212].

**Figure 4 biomedicines-10-02172-f004:**
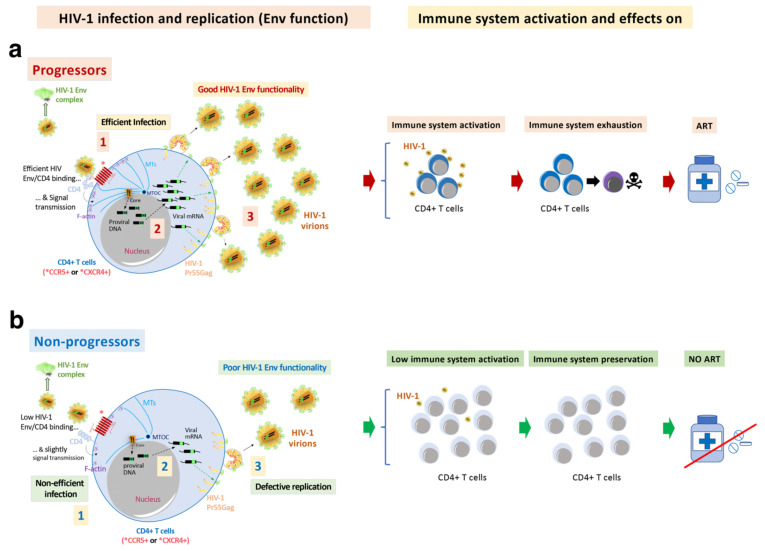
**HIV-1 Env function correlates with clinical outcome, immune system activation, exhaustion and cell damage:** (**a**) Our works indicate that HIV-1 Envs of viruses from viremic and progressors patients (i.e., Progressors, RPs and VNPs) are associated with efficient viral infection (step 1, in Progressors) and replication (step 3, in Progressors), favouring viral diversity (step 2, in Progressors) and Env gain of functionality (step 1). This gain is linked with an increasing length of the variable loops of the gp120 subunit of the Env viral complex and of N-linked glycosylation sites (PNGs) over the course of the epidemic. These functional Envs trigger cell signals activating target CD4+ T cells. Long-lasting activation leads to immune exhaustion. Progressors HIV-1 patients must follow ART, in order to avoid the development of AIDS; (**b**) On the contrary, non-functional HIV-1 Envs are associated with viruses of non-progressors HIV-1 individuals (i.e., LTNPs, LTNP-ECs and viremic LTNPs (vLTNPs)) presenting deficiencies in infection (step 1, in non-progressors), viral diversity (step 2, in non-progressors) and replication (step 3, in non-progressors). The HIV-1 *envs*’ sequences of LTNP-ECs, after thirty years of natural control of viral replication and viral pathogenesis, are close to the sequence of the T/F virus. This poor Env functionality could help the immune system to control the infection, preserving its functions. There are LTNP individuals and particularly LTNP-ECs or exceptional ECs that control HIV-1 infection for up to thirty years without any ART [44]. In this last study as well as in other investigating cases of HIV-1 functional cure, although there was not a direct analysis of Env functionality, all individuals showed undetectable VLs, and defects in viral replication and in the viral genome [44,68,213,214].

## 6. Conclusions and Perspectives

As a brief outline, HIV-1 LTNP individuals present viruses with non-functional or defective Envs correlating with controller clinical outcomes, whereas HIV-1 from viremic patients bear functional Envs and are linked to infection progression *in vivo* and pathogenesis. Therefore, viral control could be the result of the balance between virus infection efficiency and immune responses which could reasonably be more efficient against viruses with defective Envs with very limited evolution, as reported for HIV-1 Envs from LTNP individuals [44,70,71,72,73,80,120,121,127]. A consequence of this condition, as opposed to what occurs in progressors even under ART [215], is that non-functional HIV-1 Envs from LTNPs will lead to a limited activation and a lower exhaustion of the immune system [216] (Figure 4).

We have investigated mutations that could characterize the *env* proteins responsible for the deleterious phenotype. We identified three important changes in the sequences of the viruses of cluster individuals who have controlled the infection for more than twenty-five years [73]. The three mutations in the V1, C2 and V4 *env* regions, positions 140 (T140, V1), 279 (A279, C2) and 400 (T400, V4), respectively, were associated with the inefficient signaling and functionality of the Env glycoproteins of this cluster and their inefficient infectivity [121]. In the study of Pérez-Yanes *et al*. [70], we analyzed the protein sequences of the viruses from the different groups of infected individuals, and we found that the controller subjects showed shorter and less glycosylated sequences than viruses from progressors’ individuals (see Table 2 in [70]). However, all these analyses have been performed on a limited number of viral Envs, and we are in the process of investigating mutations in a wider selection of viruses.

As a final summary, in different studies in our laboratories, we have described that the HIV-1 Envs from non-controlling HIV infected individuals, like VNPs, progressors and RPs patients, are fully functional regarding the CD4-mediated pore fusion formation, viral transmission, virus entry, viral infection, a productive signaling [70,72,121,127] as well as in autophagy induction. These characteristics are in contrast with what we found in the viral Envs from vLTNPs and ECs that displayed very poor CD4 binding, fusion and viral transfer capacities resulting in low viral replication. In this regard, the VL is considered a marker for the categorization of infected Individuals [24]. Thus, the ability to control viral replication in infected individuals has important consequences on viral infection and disease progression. HIV-1 set-point viral load (VLS) has been directly associated with viral infection evolution *in vivo* and predicts disease ([12,24,217], reviewed in [1,5,218,219]). Our and other related studies have associated viral control with non-functional or functional HIV-1 Envs of viruses from LTNPs and Progressor patients, respectively.

It is important to determine whether one or several *env*/Env sequence imprints associated with Env functional loss exist. If this is the case, it will be relevant to analyze their relationship with LTNP clinical outcomes (i.e., ECs and vLTNPs) and their contribution to the elicitation of robust immune responses found in LTNP individuals and the non-progressor or controller phenotype. These questions, which remain unanswered, are important aspects that need to be addressed to make progress in the HIV-1 field. These non-functional LTNP-Envs could be potentially used to develop nAbs against functional HIV-1-Envs. 

Taken together, these studies support the hypothesis that the functionality of the viral Env is a prime characteristic determining *in vivo* viral replication control and pathogenesis. The inefficient HIV-1 Envs from HIV-1 controllers could help as potential prototypes in the investigation of new strategies for vaccination, functional cure and virus eradication.

## Figures and Tables

**Figure 1 biomedicines-10-02172-f001:**
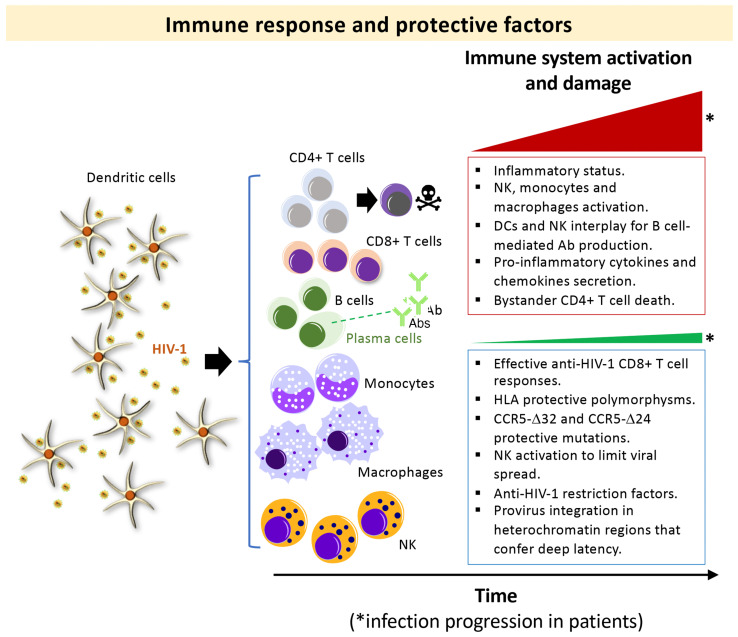
**Scheme summarizing the events in primary HIV-1 infection associated with the immune control and damage, in HIV-1 progressor and non-progressor phenotypes:** Main immune responses and damage associated with progressors HIV-1 infected individuals (*top box*) and non-progressors (*bottom box*) clinical phenotypes. *Top box*, in an acute phase of the HIV-1 infection, antigen-elicited rapid responses of innate immune cells lead to the activation of natural killer (NK) cell receptors together with monocytes/macrophages and the release of inflammatory cytokines/chemokines. This HIV-1 elicited-NK cells activation leads to secretion of IFN-γ and MIP-1β to limit viral spread [108], modulate the adaptive response in an interplay with DCs [109], and shape the induction of antibodies through the elimination of follicular T cells (Tfh) [110]. Macrophages and microglia that survive after acute HIV-1 infection could become viral reservoirs [111]. High VL is associated with predominant destruction of bystander non-infected CD4+ T mediated by HIV-1 Env (reviewed in [2,112,113]), a process we have previously reported to be dependent on the HIV-1 gp41-Env subunit by promoting autophagy and apoptosis [114,115,116,117,118,119]. *Bottom box*, some host factors, such as viral antigen-elicited CD8+ T cell response and Th1-type cytokine production [66,105,106,107], NK cell receptors [67], HLA polymorphisms [61] and CCR5 protective mutations (i.e., homo and heterozygous CCR5-Δ32 and heterozygous CCR5-Δ24 deletions/genotypes) [45,48,49,50,51,52,53,54,56] together with a limited pro-viral reservoir [68] and some restriction factors [69] have been related to a protective phenotype against HIV-1 infection. *Red and *green triangles represent how the viral infection progresses in HIV-1 infected patients of extreme different clinical phenotype: progressors (red triangle) and non-progressors (green triangle). The boxes summarized the main associated immune responses observed for the related clinical phenotype. In our studies, LTNPs, LTNP-ECs and vLTNPs individuals presented diversity in the HLA protective alleles and a lack of any protective CCR5 genotype, with the HIV-1 *env*/Env functional defect that could be the factor related to the control of clinical outcome [26,44,70,71,72,73]. Likewise, in LTNP-ECs, functional viruses would resist immune-mediated elimination through chromosomal integration into heterochromatin locations conferring deep latency and protecting against immune targeting [120].

**Figure 2 biomedicines-10-02172-f002:**
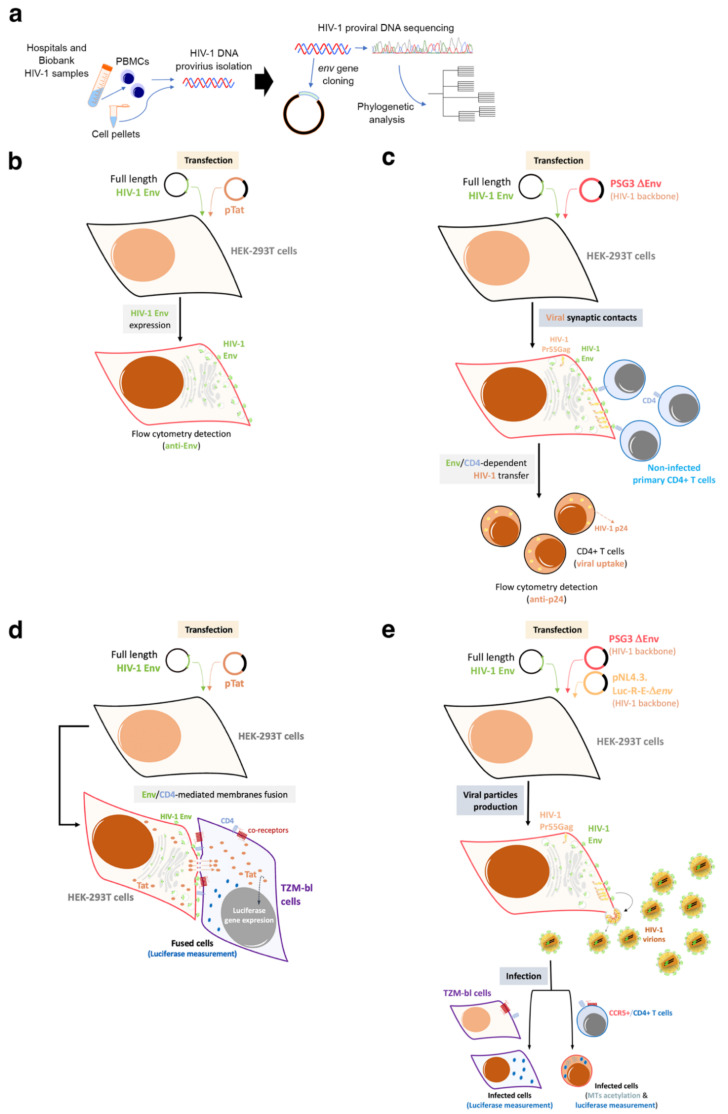
**Schemes of the experimental procedures followed to characterize primary HIV-1 *envs*/Envs from HIV-1 infected patients:** We studied viral Envs of viruses from individuals with different clinical outcomes, such as LTNP-ECs, vLTNPs, VNPs, Progressors and RPs to investigate the role of the viral Env protein in HIV-1 pathogenesis. (**a**) We cloned full-length viral *envs* in expression plasmids from the viruses of these individuals. The viral clones were completely sequenced at the nucleotide level and submitted to phylogenetic analysis. These viral *envs*/Envs were analyzed by multiple phenotypic characterizations to see the principal properties of their viruses as presented in the following panels; (**b**) Env expression: HEK-293T cells were co-transfected with a ptat Δ*env* HIV-1 expression plasmid together with reference or primary full-length viral *env* HIV-1 expression plasmid. By using the specific anti-Env antibody, cell-surface Env expression was analyzed by flow cytometry; (**c**) Env-mediated viral transfer: HEK-293T cells producing HIV-1 virions bearing reference or primary Envs were co-cultured with primary CD4+ T cells. Flow cytometry was used together with specific anti-p24 antibody to measure HIV-1 transfer to CD4+ T cells; (**d**) Env-mediated fusion activity: HEK-293T cells transfected with the *env* defective pSG3-HIV-1 backbone and primary *envs* plasmids (i.e., producing HIV-1 virions) or cells over-expressing the viral Env together with Tat viral protein (pTat construct) were co-cultured with target TZM-bl cells. Then, Env fusion capacity was measured by the magnitude of Tat-induced Luciferase activity in fused cells; (**e**) Env-mediated viral infection: TZM-bl cells were infected with serial dilutions of HIV-1 virions isolated from HEK-293T cells cotransfected with Δ*env* pSG3-HIV-1 and with primary or reference HIV-1 Envs. Infectivity capacity was determined in TZM-bl cells by measuring the luciferase activity in HIV-1 infected TZM-bl cells or in CD4+ T CEM.NKR-CCR5 cells by quantifying luciferase activity associated with the enter infectious Δ*env* pNL4-3.Luc.R-E- pseudovirus carrying primary Env.

**Figure 3 biomedicines-10-02172-f003:**
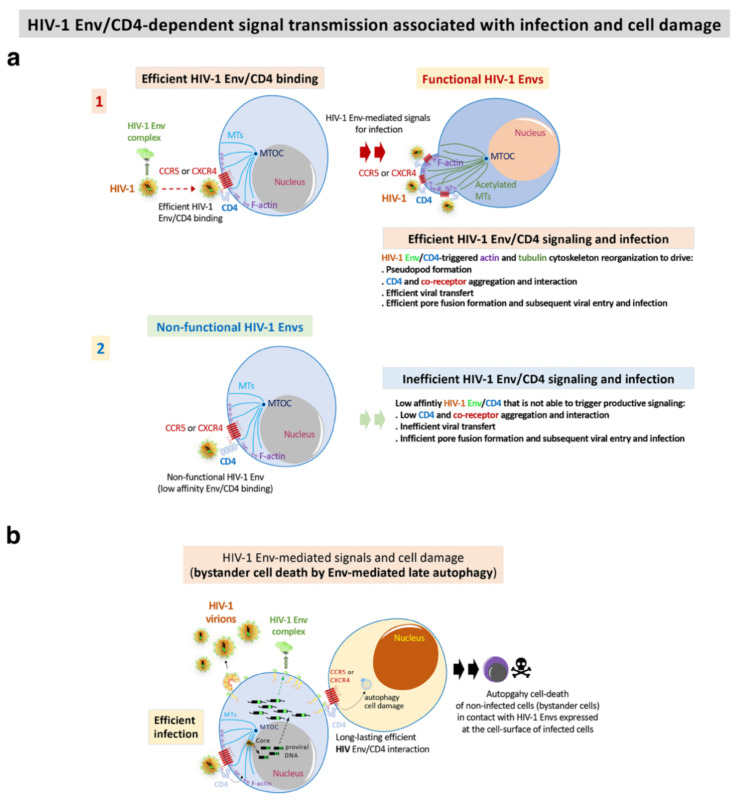
**HIV-1 Env-mediated productive cell signals and cell damage:** (**a**) functional HIV-1 Envs of viral isolates from progressors (progressors and RPs) and viremic (VNPs) patients efficiently binds to CD4 for the promotion of the F-actin and MTs reorganizations and post-translational modifications. This signal drives pseudopod formation in CD4+ T cells where CD4 and chemokine co-receptors for HIV-1 infection reorganize, aggregate and interact (step 1) [70,72,121,123,125,126,140,142]. These events are required for efficient pore fusion formation, viral transfer, virus entry and infection. On the contrary, non-functional HIV-1 Envs of viral isolates from non-progressors’ patients (LTNP-ECs, LTNPs and vLTNPs) are not able to bind to CD4 with high affinity, and thereby are unable to reorganize the cytoskeleton and favour all the events for productive viral transfer and infection (step 2) [70,72,121]; (**b**) functional HIV-1 Envs of viral isolates from viremic and progressors (VNPs and RPs) patients efficiently bind to CD4 to trigger late autophagy with subsequent cell death of non-infected CD4+ T cells (bystander cells) by contact [72].

## Data Availability

References for this review were identified through searches of Pub- Med for published articles and from our publications.

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
