# Peer review of "Contribution of the HIV-1 Envelope Glycoprotein to AIDS Pathogenesis and Clinical Progression"

_biomedicines, 2022, doi:10.3390/biomedicines10092172_

Round 1
Reviewer 1 Report
The manuscript presented by Valenzuela-Fernández et al. is a review about the implication of HIV-1 Env protein in the progression or hindering of the infection. The authors are writing the review mainly based on their previously conducted and published studies, highlighting their important findings related to the topic. While the topic is interesting and deserves thorough research, I found the manuscript a little hard to read and follow, and I wish that it was more well written. My first impression is that the title was a little misleading, since the review does not really speak about the pathophysiology of Env in infection, or even clinical progression to AIDS (a clinical spectrum). From what I have read, the authors are reporting majorly on their findings on the linkage between Env and viremia, which is not the same as pathophysiology of infection and clinical progression to AIDS.
The review needs thorough revision in terms of English language, since it contains many grammatical errors, such as but not limited to:
54: primary infection
91: apart from
94: (Env) have “attracted” numerous works
126: focused on..
128: isolated viral sequences from “infected individuals”
143: based on
194: in addition to
253: . Fully functional HIV-1 Envs are linked “to” viremic and progressor’s clinical ..
446-onwards: needs re-writing as it is currently un-understandable in its current form.
…etc
.
There is also another issue with the figures, figure 2 for example is an exact copy of another figure that the authors used in a previous publication, which is a potential problem.
The concluding remarks section is too long and also contains a figure which is highly unusual, maybe rename the section? or add a discussion section perhaps?
Author Response
Point-by-point reply to Reviewer#1:
The manuscript presented by Valenzuela-Fernández et al. is a review about the implication of HIV-1 Env protein in the progression or hindering of the infection. The authors are writing the review mainly based on their previously conducted and published studies, highlighting their important findings related to the topic. While the topic is interesting and deserves thorough research, I found the manuscript a little hard to read and follow, and I wish that it was more well written.
We would like to thank Reviewer #1 for these comments about the manuscript-Review.
The revised version of the manuscript has been English language edited by an experienced English language editor of scientific manuscripts for international publications. We have acknowledged this in the Acknowledgments section of the Ms (page 14, lines 576-578).
My first impression is that the title was a little misleading, since the review does not really speak about the pathophysiology of Env in infection, or even clinical progression to AIDS (a clinical spectrum). From what I have read, the authors are reporting majorly on their findings on the linkage between Env and viremia, which is not the same as pathophysiology of infection and clinical progression to AIDS.
The referee is correct in that we have not performed a direct study on the pathophysiology of infection but an analysis on the role of the viral envelope in viral replication. However, as stated a long time ago, viral load is the best marker for the categorization of infected individuals as shown by John Mellors et al. (Mellors JW, et al. Prognosis in HIV-1 infection predicted by the quantity of virus in plasma. Science. 1996 May 24;272(5265):1167-70. doi: 10.1126/science.272.5265.1167. Erratum in: Science 1997 Jan 3;275(5296):14). Following this line, we assumed that the control or lack of control in viral replication in infected individuals has important consequences on viral Infection, and the viral load at the set point has been directly associated with the viral infection evolution in vivo and predicts disease outcome (Kelley CF, et al. The relation between symptoms, viral load, and viral load set point in primary HIV infection. J Acquir Immune Defic Syndr. 2007 Aug 1;45(4):445-8. doi: 10.1097/QAI.0b013e318074ef6e; Mellors JW, et al. Prognosis in HIV-1 infection predicted by the quantity of virus in plasma. Science. 1996 May 24;272(5265):1167-70. doi: 10.1126/science.272.5265.1167. Erratum in: Science 1997 Jan 3;275(5296):14; Deeks SG and Walker BD. Human immunodeficiency virus controllers: mechanisms of durable virus control in the absence of antiretroviral therapy. Immunity. 2007 Sep;27(3):406-16. doi: 10.1016/j.immuni.2007.08.010; Sáez-Cirión A, et al. ANRS VISCONTI Study Group. Post-treatment HIV-1 controllers with a long-term virological remission after the interruption of early initiated antiretroviral therapy ANRS VISCONTI Study. PLoS Pathog. 2013 Mar;9(3):e1003211. doi: 10.1371/journal.ppat.1003211; Hughes JP, et al. Partners in Prevention HSV/HIV Transmission Study Team. Determinants of per-coital-act HIV-1 infectivity among African HIV-1-serodiscordant couples. J Infect Dis. 2012 Feb 1;205(3):358-65. doi: 10.1093/infdis/jir747; Bruyand M, et al. Groupe d'Epidémiologie Clinique du SIDA en Aquitaine (GECSA). Role of uncontrolled HIV RNA level and immunodeficiency in the occurrence of malignancy in HIV-infected patients during the combination antiretroviral therapy era: Agence Nationale de Recherche sur le Sida (ANRS) CO3 Aquitaine Cohort. Clin Infect Dis. 2009 Oct 1;49(7):1109-16. doi: 10.1086/605594. Reviewed in Deeks SG, et al. HIV infection. Nat Rev Dis Primers. 2015 Oct 1;1:15035. doi: 10.1038/nrdp.2015.35; Patel P, et al. Estimating per-act HIV transmission risk: a systematic review. AIDS. 2014 Jun 19;28(10):1509-19. doi: 10.1097/QAD.0000000000000298; Simon V, et al. HIV/AIDS epidemiology, pathogenesis, prevention, and treatment. Lancet. 2006 Aug 5;368(9534):489-504. doi: 10.1016/S0140-6736(06)69157-5; Gonzalo-Gil E, et al. Mechanisms of Virologic Control and Clinical Characteristics of HIV+ Elite/Viremic Controllers. Yale J Biol Med. 2017 Jun 23;90(2):245-259).
To make more clear this association, we added a new paragraph explaining the correlation of viral load with HIV-1 disease progression, as follows (page 13, lines 525-532):
“In this regard, the viral load is considered a marker for the categorization of infected Individuals [24]. Thus, the ability to control viral replication in infected individuals has important consequences on viral infection and disease progression. HIV-1 set-point viral load (VLS) has been directly associated with viral infection evolution in vivo and predicts disease ([12,24,217], reviewed in [1,5,218,219]). Our and other related studies have as-sociated viral control with non-functional or functional HIV-1 Envs of viruses from LTNPs and Progressor patients, respectively.”
Furthermore, to approach the correlation of viral replication with disease progression, we functionally characterized HIV-1 Envs from virus of patients with different clinical phenotypes. We analyzed the linkage of Env functionality with the respective clinical phenotypes (progressors or non-progressors). Fully functional viral proteins have been described in the envelopes from non-controller patients that progress clinically; in contrast we observed that the lack of viral replication characteristic found in controller individuals (vLTNPs and EC) was linked with the lack of functionality of the viral envelopes.
Therefore, we prefer to maintain the proposed title of the manuscript.
The review needs thorough revision in terms of English language, since it contains many grammatical errors, such as but not limited to:
As suggested by Reviewer #1, the revised version of the manuscript has been English language edited by an experienced English language editor of scientific manuscripts for international publications. We have acknowledged this fact in the Acknowledgments section of the Ms (page 14, lines 576-578).
We would like to sincerely thank Reviewer #1 for all the proposed corrections listed below:
54: primary infection; OK.
91: apart from; OK.
94: (Env) have “attracted” numerous works; OK.
126: focused on.. ; OK.
128: isolated viral sequences from “infected individuals”
(we adapted the sentence as follows: The isolated viral env sequences (full-length viral env) from infected individuals were cloned into expression plasmids).
143: based on; OK.
194: in addition to; OK.
253: . Fully functional HIV-1 Envs are linked “to” viremic and progressor’s clinical . ; OK.
446-onwards: needs re-writing as it is currently un-understandable in its current form.
...etc.
We would like to thank Reviewer #1 for this remark. As suggested by Reviewer #1, the revised version of the manuscript has been English language edited by an experienced English language editor of scientific manuscripts for international publications.
Moreover, we propose a “summary conclusions” in the new section 6 (Concluding remarks and perspectives) (pages 13; lines 519-546), as follows:
“As a final summary, in different studies in our laboratories, we have described that the HIV-1 Envs from non-controlling HIV infected individuals like VNPs, progressors and RPs patients are fully functional regarding the CD4-mediated pore fusion formation, viral transmission, virus entry, viral infection, a productive signaling [70,72,121,127] as well as in autophagy induction. These characteristics are in contrast with what we found in the viral Envs from vLTNPs and ECs that displayed very poor CD4 binding, fusion and viral transfer capacities resulting in low viral replication. In this regard, the viral load is considered a marker for the categorization of infected Individuals [24]. Thus, the ability to control viral replication in infected individuals has important consequences on viral infection and disease progression. HIV-1 set-point viral load (VLS) has been directly associated with viral infection evolution in vivo and predicts disease ([12,24,217], reviewed in [1,5,218,219]). Our and other related studies have associated viral control with non-functional or functional HIV-1 Envs of viruses from LTNPs and Progressor patients, respectively.
It is important to determine whether one or several env/Env sequence imprints as-sociated with Env functional loss exist. If this is the case, it will be relevant to analyze their relationship with LTNP clinical outcomes (i.e., ECs and vLTNPs) and their contribution to the elicitation of robust immune responses found in LTNP individuals and the non-progressor or controller phenotype. These questions, which remain unanswered, are important aspects that need to be addressed to make progress in the HIV-1 field. These non-functional LTNP-Envs could be potentially used to develop nAbs against functional HIV-1-Envs.
Taken together, these studies support the hypothesis that the functionality of the viral Env is a prime characteristic determining in vivo viral replication control and pathogenesis. The inefficient HIV-1 Envs from HIV-1 controllers could help as potential prototypes in the investigation of new strategies for vaccination, functional cure and virus eradication.”
There is also another issue with the figures, figure 2 for example is an exact copy of another figure that the authors used in a previous publication, which is a potential problem.
We would like to thank Reviewer #1 for this remark.
Figure 2 is quite similar to a previous figure of our published work but it is not an exact copy in all panels. Panel A is completely original and panel D is different containing new information, a new model for CD4+ T cell infection.
These represent the experiential strategies that could be used to functionally characterize viral HIV-1 Envs, therefore the schemes should be conserved. However, as suggested by the Reviewer, we have modified the aspects of packaging HEK-293T cells and permissive TZM-bl cells in the new revised Figure 2, in order to avoid any potential problem.
The concluding remarks section is too long and also contains a figure which is highly unusual, maybe rename the section? or add a discussion section perhaps?
We would like to thank again Reviewer #1 for this key remark.
We have modified section 5 with two new subheadings: “5. Discussion” (page 10, line 395), including Figure 4, and final “6. Concluding remarks and perspectives” (page 13, line 496), where we have also “unanswered questions in the field and/or any unresolved controversies”, as proposed by Reviewer #2.
We would like to sincerely thank Reviewer #1 for helping us improve the manuscript.

Reviewer 2 Report
This is an excellent review of the work that the authors (and others) have done in the field of HIV-1 Env-mediated cytotoxicity. They did a nice job bringing out the connection between gp41 function and cell death, particularly bystander cell death. While the piece was very informative, I might have liked to see a discussion on determinants in Env that are linked to cytopathicity and disease progression, e.g. from the sequence information, perhaps with a table.
There are a few phrases that are unusual: "ineffective viral proteins" - line 35; "high-repercussion study" - line 84; "plentiful works" - line 89; "concentrated numerous works" - line 92; "different works" - line 96. By "works" I assume the authors mean "studies."
I was a bit confused by line 101 in the legend to Figure 1 where the authors mention that innate immune cells are "involved in viral spread limitation." The red box seems to focus on effects and responses that promote virus replication and disease progression. That made me wonder what the green triangles are and red triangles in the figure are meant to represent.
I suggest the authors finish the piece with a paragraph describing the unanswered questions in the field and/or any unresolved controversies.
Author Response
Point-by-point reply to Reviewer#2:
This is an excellent review of the work that the authors (and others) have done in the field of HIV-1 Env-mediated cytotoxicity. They did a nice job bringing out the connection between gp41 function and cell death, particularly bystander cell death. While the piece was very informative, I might have liked to see a discussion on determinants in Env that are linked to cytopathicity and disease progression, e.g. from the sequence information, perhaps with a table.
We would like to thank Reviewer #2 for these kind comments about the manuscript-Review.
Concerning the remark made by Reviewer #2 about “a discussion on determinants in Env that are linked to cytopathicity and disease progression, e.g. from the sequence information, perhaps with a table.”, we consider that the last paragraph in the revised version of the manuscript addresses the remark made by the Reviewer: “the unanswered questions in the field and/or any unresolved controversies”. Specifically, HIV-1 Env sequence information and its relationship with each Env function and the viral infection progression in vivo represent part of our research in progress and key questions to be solved.
Thus, we have modified section 5 with two new subheadings: “5. Discussion” (page 10, line 395), including Figure 4, and final “6. Concluding remarks and perspectives” (page 13, line 496), where we have addressed “unanswered questions in the field and/or any unresolved controversies”, as proposed by Reviewer #2, and have introduced the following paragraph concerning env/Env sequence information (page 13, lines 507-518):
“We have investigated mutations that could characterize the env proteins responsible for the deleterious phenotype. We identified three important changes in the sequences of the viruses of cluster individuals who have controlled the infection for more than twenty-five years [73]. The three mutations in the V1, C2, and V4 env regions, positions 140 (T140, V1), 279 (A279, C2) and 400 (T400, V4), respectively were associated with the inefficient signaling and functionality of the Env glycoproteins of this cluster and their inefficient infectivity [121]. In the study of Pérez-Yanes et al., [70], we analyzed the protein sequences of the viruses from the different groups of infected individuals and we found that the controller subjects showed shorter and less glycosylated sequences than viruses from progressors individuals (see Table 2 in [70]). However, all these analyses have been performed on a limited number of viral Envs and we are in the process of investigating mutations in a wider selection of viruses.”
There are a few phrases that are unusual:
We would like to mention that the revised version of the manuscript has been English language edited by an experienced English language editor of scientific manuscripts for international publications. We have acknowledged this fact in the Acknowledgments section of the Ms (page 14, lines 576-578).
-"ineffective viral proteins", line 35;
Changed for “poorly-functional viral proteins”
"high-repercussion study”, line 84;
Changed for “but influential study”
- line 89; "plentiful works"
Changed for “has been extensively studied”
- line 92; "concentrated numerous works"
Changed for “have attracted numerous studies”
- line 96; "different works". By "works" I assume the authors mean "studies."
Changed for “several studies”
Yes, we mean studies.
I was a bit confused by line 101 in the legend to Figure 1 where the authors mention that innate immune cells are "involved in viral spread limitation."
We agree with Reviewer #2 that this part of the sentence could be confusing or unnecessary, since the first part of the following sentence finished with “to limit viral spread”. We have removed “are involved in viral spread limitation " from the first sentence, adapting this sentence and the beginning of the following sentence, as follows (page 3, lines 99-102):
“Top box, in acute phase of the HIV-1 infection, antigen-elicited rapid responses of innate immune cells lead to the activation of natural killer (NK) cell receptors together with monocytes/macrophages and the release of inflammatory cytokines/chemokines.”
The red box seems to focus on effects and responses that promote virus replication and disease progression. That made me wonder what the green triangles are and red triangles in the figure are meant to represent.
We would like to thank you Reviewer #2 for this key remark.
Red and green triangles represent how the viral infection progresses in HIV-1 infected patients of extreme, different clinical phenotype: progressors (red triangle) and non-progressors (green triangle). The box below each triangle summarizes the main associated immune responses observed for the clinal phenotypes. This idea is now more clearly indicated in the figure legend to Figure 1 (page 3), as follows:
(page 3, lines 98-99)
“Main immune responses and damage associated with progressors HIV-1 infected individuals (top box) and non-progressors (bottom box) clinical phenotypes.”
(pages 3-4, lines 113-117)
“ *Red and *green triangles represent how the viral infection progresses in HIV-1 infected patients of extreme different clinical phenotype: progressors (red triangle) and non-progressors (green triangle). The boxes summarized the main associated immune responses observed for the related clinical phenotype.“
I suggest the authors finish the piece with a paragraph describing the unanswered questions in the field and/or any unresolved controversies.
We would like to thank you Reviewer #2 for this key remark.
We have modified section 5 with two new subheadings: “5. Discussion” (page 10, line 395), including Figure 4, and final “6. Concluding remarks and perspectives” (page 13, line 496), where we have addressed “unanswered questions in the field and/or any unresolved controversies”, as follows (pages 13, lines 534-546):
“It is important to determine whether one or several env/Env sequence imprints as-sociated with Env functional loss exist. If this is the case, it will be relevant to analyze their relationship with LTNP clinical outcomes (i.e., ECs and vLTNPs) and their contribution to the elicitation of robust immune responses found in LTNP individuals and the non-progressor or controller phenotype. These questions, which remain unanswered, are important aspects that need to be addressed to make progress in the HIV-1 field. These non-functional LTNP-Envs could be potentially used to develop nAbs against functional HIV-1-Envs.
Taken together, these studies support the hypothesis that the functionality of the viral Env is a prime characteristic determining in vivo viral replication control and pathogenesis. The inefficient HIV-1 Envs from HIV-1 controllers could help as potential prototypes in the investigation of new strategies for vaccination, functional cure and virus eradication.”
We would like to sincerely thank Reviewer #2 for helping us improve the manuscript.

Round 2
Reviewer 1 Report
The authors have implemented the necessary changes and improved their review.